# OpenReview forum: "Explainable Multi-modal Time Series Prediction with LLM-in-the-Loop"
_ICML.cc/2025/Conference — Submitted to ICML 2025_

### Official Review · Reviewer_soJm · 2025-03-01

**Overall Recommendation:** 3

**Summary:**

This paper proposed a multi-modal prediction framework that integrates a prototype-based time series encoder with three collaborating LLMs to deliver more accurate predictions and interpretable explanations. The closed-loop workflow – prediction, critique, and refinement – continuously boosts the framework’s performance and interpretability. Empirical evaluations demonstrate good performance.

**Claims And Evidence:**

Yes.

**Essential References Not Discussed:**

No.

**Experimental Designs Or Analyses:**

Yes.

**Methods And Evaluation Criteria:**

Yes.

**Other Comments Or Suggestions:**

See questions below.

**Other Strengths And Weaknesses:**

**Strength**: \
(1) Propose a prototype-based encoder that combines time series data with textual context, producing transparent, case-based rationales. \
(2) Detailed related works. \
(3) Comprehensive comparative experiments were conducted to meticulously evaluate and analyze the performance. \
(4) The writing is clear and the method pipeline is easy to follow.

**Weaknesses**: \
(1) In my opinion, this paper lacks sufficient innovation. \
(2) The related work section, especially for LLMs for time series analysis, merely lists various methods without comparing their strengths and weaknesses, failing to provide a clear motivation, or starting point for the proposed approach in this paper. \
(3) The experimental section lacks thoroughness and completeness. For instance, it fails to evaluate the model's performance on future event time prediction tasks, omits comparisons with advanced temporal point process models, and does not explore the impact of different base LLM models.

**Questions For Authors:**

(1) I find that the method proposed in this paper appears to be a mere amalgamation of existing tools, which leads me to challenge its novelty. I think the paper lacks sufficient innovation. But I am interested in seeing how other reviewers assess the novelty of this work. \
(2) How to divide text data into multiple meaningful segment? Please explain more about the text data and provide examples for clarity. \
(3) This paper primarily focuses on predicting discrete labels, which underutilizes the strengths and flexibility of LLM. Could this model be adapted to predict future event times? If so, what extensions would be necessary? \
(4) For sequence data, temporal point process model is a good choice to describe dynamics. I suggest the authors to compare the model performance with advanced TPP models. \
(5) Could the authors compare the performance of different base LLM models and provide a corresponding reproducibility analysis? \
(7) Could the authors provide detailed prompt designs for LLM models?

**Relation To Broader Scientific Literature:**

LLM, multi-modal time series analysis, time series explanation.

**Theoretical Claims:**

Yes.

---

> ### Author Rebuttal · Authors · 2025-04-01
>
> **Question 1 and Weakness 1,2: Innovation and scope:**
>
> We appreciate the reviewer’s comments and the opportunity to clarify our contributions. Our work is motivated by the need for effective explanation, multi-modal time series understanding, and contextual reasoning—areas that are underexplored in current literature. Compared with existing LLM-based methods, our approach offers a unique advantage through: (1) explicit, case-based explanations grounded in multi-modal time series inputs, and (2) closed-loop interactions between time series models and LLMs that enable iterative understanding and refinement of real-world contexts. We will revise the related work section to more clearly articulate these distinctions.
>
> **Question 2:  Meaningful text segment**:
>
> Thanks for the question! As discussed in Section 3.2.1 (Page 3), capturing meaningful text segments (i.e, text prototypes) relies on two components, the pretrained language model producing text embeddings for input texts, and the sequence encoder mapping text embeddings and performing prototype learning. 1) The first component decides the granularity of prototypes at the input level. For example, Bert represents each text input as multiple tokens (token-level embeddings), while S-bert represents it as multiple sentences (sentence-level embeddings). 2) The second component further encodes these text embeddings. We use the convolution-based encoder to get segment representations by applying sliding windows over the text embeddings. With multiple consecutive segment representations, we perform prototype learning with regularizations and projections to identify the most typical segments.
>
> **Question 3 & 4 and Weakness 3:**
>
> **Event time prediction and temporal point process:** Thanks for the question! We agree with the reviewer that this is an interesting extension. However, we would like to clarify that our setting is fundamentally different from the temporal point process (TPP) setting and its event time prediction task.  TPP models are specifically designed for **irregular event sequences**, but our problem setting and datasets are for **regularly sampled multivariate time series** with categorical or continuous outcomes at each step, following the standard time series prediction paradigm. Our objective is not to predict when the next event will occur, but rather to predict what will happen at each regularly observed time step. Therefore, event-time prediction is not applicable in our setting, and TPP models are not directly relevant for comparison.
>
> Adapting our method to this task would require major changes: the encoder must handle irregular timesteps, and the output layer must predict inter-event intervals or model intensity functions. While our prototype design and LLM agents can be modified, it is outside the scope of the current work. Nevertheless, we appreciate the reviewer’s insight and agree that TPP is a valuable framework for modeling temporal dynamics, and will discuss representative works and their differences in the updated version.
>
> **Beyond discrete label prediction:** We provided regression results in Appendix F (pages 23-25). We also provide baselines comparisons and model analysis to demonstrate its efficacy for regression tasks. Please refer to the rebuttal for **reviewer g9bw** for detailed results and discussions.
>
> **Question 5 and Weakness 3: Base LLMs**
>
> Thank you for this question! To explore the effect of different base LLMs (Gemini-2.0 Flash and GPT-4o-mini), we provide the experiment results and analysis on Healthcare (Test-Positive) data. For fair comparisons, we used the same prompts and ran the same number of iterations and followed the same settings detailed in Appendix A.2 and A.4 (pages 13-14). We still use the same temperature setting (0.7 for content generation, 0.3 for prediction), as it yields the best performance empirically. The results are shown below.
>
> |Base LLM|F1|AUC|
> |-|-|-|
> |GPT-4o-mini|0.932|0.981|
> |Gemini-2.0-Flash|0.937|0.983|
> |GPT-4o, default|0.987|0.996|
>
> GPT-4o clearly outperforms both GPT-4o-mini and Gemini-2.0-Flash, due to its better reasoning and contextual understanding capabilities (larger model size & pre-training corpus). Both GPT-4o-mini and Gemini-2.0-Flash are still competitive compared with baselines listed in Table 1, Page 7. It reveals the impact of the LLM capability on the effectiveness of our framework, especially in tasks demanding real-world context understanding.
>
> We also provide the plot of iterative analysis in [this link](https://anonymous.4open.science/r/rebuttal-D4F5/Base-LLMs-iteration.pdf), where we can also observe the performance improvement over iterations for different base LLMs.
>
> **Question 7: Detailed prompt designs**
>
> As indicated in Section 3.3.2 (Page 5), we provided the detailed prompt templates in Figures 13-17 in Appendix D (Pages 18-21). We also provided the code containing specific prompts in the supplementary materials. Please refer to them.

---

### Official Review · Reviewer_g9bw · 2025-03-11

**Overall Recommendation:** 3

**Summary:**

The paper introduces TimeXL, a multi-modal prediction framework designed to integrate both time series data and textual information, addressing a common limitation in existing time series models that often neglect auxiliary textual data available in real-world scenarios. A key contribution of the paper is a new encoding approach for textual data, leveraging prototypical explanations to extract meaningful representations. Building on this, the framework employs iteratively three agents that progressively refine the textual data, modifying it in a structured manner to improve the overall prediction quality. The effectiveness of TimeXL is demonstrated empirically in Table 1 and the appendix, where the authors present experimental results showcasing the superiority of their approach over existing methods.

**Claims And Evidence:**

The claims made in the submission are generally well-supported by clear and convincing evidence. The authors provide strong quantitative results that demonstrate the superiority of their approach, and they supplement this with qualitative examples that illustrate how the data is transformed.

**Essential References Not Discussed:**

--

**Experimental Designs Or Analyses:**

Yes, I reviewed the experimental design and analyses presented in the submission. The comparisons against 16 baselines across three domains (health, finance, and weather) appear sound and provide a comprehensive evaluation of the proposed method. The quantitative results are compelling, and the qualitative examples help illustrate the data transformations. One potential improvement could be the inclusion of additional evaluation metrics, such as MSE or MAE, to provide more granular insights at each time step

**Methods And Evaluation Criteria:**

The proposed methods and evaluation criteria are well-suited to the problem at hand. The authors conduct a thorough evaluation by comparing their approach to 16 baselines across three distinct domains—health, finance, and weather. This diverse benchmarking provides a strong basis for assessing the generalizability and effectiveness of their method.

One limitation is the choice of evaluation metrics. While the focus on binary classification (e.g., rain/no-rain) is relevant, it would be beneficial to also report MSE or MAE, as these are commonly used metrics that provide insight into performance at each time step. Including such results would strengthen the evidence presented.

**Other Comments Or Suggestions:**

--

**Other Strengths And Weaknesses:**

--

**Questions For Authors:**

--

**Relation To Broader Scientific Literature:**

The paper's key contributions relate to broader scientific literature in several important ways:

* **Text as additional data beyond time series** - Extends traditional numerical forecasting by elevating text to a primary data source rather than supplemental features

* **Agenic AI application** - Evolves from passive prediction systems to active forecasting agents that autonomously direct information gathering

* **Explanations for text encoding** - Advances beyond post-hoc explanations by integrating interpretability directly into the encoding process

* **Iterative text refinement** - Connects to active learning approaches but specifically for text improvement, introducing dynamic feedback loops absent in static-input forecasting systems.

Together, these contributions form a cohesive framework addressing limitations in forecasting system interpretability, adaptability, and information utilization.

**Theoretical Claims:**

There are no theoretical claims in the paper

---

> ### Author Rebuttal · Authors · 2025-04-01
>
> **The regression-based setting:**
>
> We sincerely appreciate the reviewer for the comments. In Appendix F (Pages 23-25), we implemented a regression-based variant and provided a demonstration on the same finance dataset with numerical ground truths to show its capability for numerical value forecasting. To further address the reviewer's concern, we provide more details and additional experiments below.
>
> As introduced in Appendix F,  we adapt TimeXL for regression with two minor modifications. First, we add a regression branch in the encoder design, as shown in Figure 23. On the output of time series prototype layers, we reversely ensemble each time series segment representation as a weighted sum of time series prototypes, and add a regression head for prediction. Accordingly, we add another regression loss term to the learning objectives. Second, we adjust the prompt for prediction LLM by adding time series inputs and requesting numerical forecasts, as shown in Figure 24. As such, TimeXL is equipped with regression capability. Next, we evaluate our method on the same finance dataset with the same task as classification, despite that the main prediction target is the raw material stock price instead of trends. Here we follow the same settings detailed in section 4.1, and Appendix A.2-A.4. We update the results by comparing state-of-the-art baselines in the table  below..
>
> | **Model**      | **RMSE** | **MAE** | **MAPE(%)** |
> |-|-|-|-|
> | **DLinear**        | 7.871    | 6.400   | 4.727       |
> | **Autoformer**   | 7.215    | 5.680   | 4.263       |
> | **Crossformer**  | 7.205    | 5.313   | 3.808       |
> | **TimesNet**  | 6.978    | 4.928   | 3.512       |
> | **iTransformer**  | 5.877    | 4.023   | 2.863       |
> | **TSMixer**       | 7.447    | 5.509   | 3.911       |
> | **FreTS**  | 7.098    | 4.886   | 3.460       |
> | **PatchTST**    | 5.676    | 4.042   | 2.853       |
> | **LLMTime**    | 11.545   | 5.300   | 3.774       |
> | **PromptCast**      | 4.728    | 3.227   | 2.306       |
> | **OFA**        | 6.906    | 4.862   | 3.463       |
> | **Time-LLM**        | 6.396    | 4.534   | 3.238       |
> | **TimeCMA**        | 7.187    | 5.083   | 3.620       |
> | **MM-iTransformer**  | 5.454    | 3.789   | 2.687       |
> | **MM-PatchTST**    | 5.117    | 3.493   | 2.491       |
> | **TimeCAP**  | 4.456   | 3.088   | 2.196      |
> | **TimeXL**  | **4.161** | **2.844** | **2.035** |
>
> The main observations are consistent with the classification setting: The multi-modal variants of state-of-the-art baselines (MM-iTransformer and MM-PatchTST) benefit from incorporating real-world contexts; Our method achieves the best results, highlighting the advantage of synergizing multi-modal time series encoder with language agents to enhance interpretability and thus predictive performance.
>
> Moreover, we provide the component analysis based on the refined texts that achieve the best validation performance across iterations. It includes evaluation results of the multi-modal time series encoder, input ablations for LLM-based predictions, and the fusion of both components (TimeXL). The results are summarized in the table below, which also show patterns consistent with those observed in the classification setting. (1) The results clearly demonstrate that real-world financial texts provide complementary information to the LLM, leading to improved accuracy in numerical prediction.  (2) The identified prototypes provide contextual guidance to prediction LLM, leading to clear performance gains.  (3) By fusing the predictions from both encoder and prediction LLM, TimeXL further improves the prediction and outperforms all variants, underscoring the effectiveness of mutual enhancement between the two components.
>
> In general, (1) highlights the importance of multi-modal inputs, while (2) and (3) highlight our proposed interaction between encoder and LLM for more accurate numerical predictions.
>
> | Variants                   | RMSE  | MAE   | MAPE(%) |
> |-|-|-|-|
> | Multi-modal Encoder       | 4.198 | 2.891 | 2.064   |
> | Prediction LLM using Time Series       | 4.728 | 3.227 | 2.306   |
> | Prediction LLM using Time Series + Text    | 4.600 | 3.121 | 2.226   |
> | Prediction LLM using Time Series + Text + Prototype (ours)   | 4.352 | 3.003 | 2.165   |
> | **TimeXL**                | **4.161** | **2.844** | **2.035** |
>
> We also provide an iteration analysis to show the effectiveness of reflection and refinement LLMs, as shown in the table below. The prediction performance quickly improves and stabilizes over iterations, which underscores the alternation steps between predictions and reflective refinements.
>
> | Iteration      | RMSE  | MAE   | MAPE(%) |
> |-|-|-|-|
> |Original | 4.344 | 2.951 | 2.103 |
> |1| 4.224 |  2.883 | 2.069 |
> |2| 4.161 |  2.844 | 2.035 |
> |3| 4.174 |  2.849 | 2.036 |

---

> > ### Comment · Reviewer_g9bw · 2025-04-07
> >
> > Thank you for your response. I’ll keep the accept score as is

---

### Official Review · Reviewer_KxU3 · 2025-03-14

**Overall Recommendation:** 3

**Summary:**

In this paper, the authors proposed a new Multi-model time series prediction model that uses prototype-based encoder with 3 LLMs to predict, reflect and refine the semantic information. Experiments on real-world datasets show the effectiveness of the proposed method.

**Claims And Evidence:**

Experiments are not enough.
1. The ablation study needs improvement to show the essential of introducing 3 LLMs.
2. The majority of the comparison methods are designed for regression tasks, which may not be suitable for studied tasks.

**Essential References Not Discussed:**

Some recent refs regarding aligning time series understanding and reasoning [1][2][3] should be discussed.
[1] From News to Forecast: Integrating Event Analysis in LLM-Based Time Series Forecasting with Reflection. NeurIPS 2024
[2] ChatTime: A Unified Multimodal Time Series Foundation Model Bridging Numerical and Textual Data. AAAI 2025
[3] ChatTS: Aligning Time Series with LLMs via Synthetic Data for Enhanced Understanding and Reasoning.
The authors cited [1][2] and missed [3]. They did not explain the difference between the proposed methods with [1] and [2], neither compared with them.

**Experimental Designs Or Analyses:**

1. The ablation study needs improvement to show the essential of introducing 3 LLMs.
2. The majority of the comparison methods are for regression tasks. It seems to be an unfair comparison, since we do not know whether the baseline methods are well-tuned for classification tasks.

**Methods And Evaluation Criteria:**

Yes

**Other Comments Or Suggestions:**

No

**Other Strengths And Weaknesses:**

Strengths:
1. The idea of using LLM to refine the input text to encoder is interesting.
2. Experiments show the strong performance of proposed methods.
Weaknesses:
1. It is difficult to tell which part contributes most to the final results, prototype or LLM loop. More ablation studies on different modules of the proposed method are needed.
2. The proposed structure is very complex and it is a bit difficult to follow the paper.

**Questions For Authors:**

1. The majority of the comparison methods are for regression tasks. However, the authors only conducted classification tasks? Why not regression tasks?
2. What is the model choice for a time series encoder?
3. How did the author perform an ablation study in Table 2? From Figure 2, we can see that the text input can be refined by LLMs. Why is the text and prototype only used for ablation study of LLM not encoder? Encoder uses prototypes and text.
4. From Algorithm 1, it seems that iterative process only goes in training?

**Relation To Broader Scientific Literature:**

The novelty of paper seems to be using prototypes to enhance the multi-modal time series prediction and using LLM to refine the input text.

**Theoretical Claims:**

No theoretical claims

---

> ### Author Rebuttal · Authors · 2025-04-01
>
> We sincerely thank the reviewer for the insightful feedback and provide our response below.
>
> **Q1 and Weakness 2: Regression tasks**
>
> We appreciate the reviewer’s interest in understanding how our approach performs on regression tasks. Below are the results of our method and state-of-the-art baselines on the same finance dataset in our paper, but with numerical ground truths.
>
> |Model|RMSE|MAE|MAPE(%)|
> |-|-|-|-|
> |DLinear |7.871|6.400|4.727|
> |Autoformer|7.215|5.680|4.263|
> |Crossformer|7.205|5.313|3.808|
> |TimesNet|6.978|4.928|3.512|
> |iTransformer|5.877|4.023|2.863|
> |TSMixer|7.447|5.509|3.911|
> |FreTS|7.098|4.886|3.460|
> |PatchTST|5.676|4.042|2.853|
> |LLMTime|11.545|5.300|3.774|
> |PromptCast|4.728|3.227|2.306|
> |OFA|6.906|4.862|3.463|
> |Time-LLM|6.396|4.534|3.238|
> |TimeCMA|7.187|5.083|3.620|
> |MM-iTransformer|5.454| 3.789|2.687|
> |MM-PatchTST|5.117|3.493|2.491|
> |TimeCAP|4.456|3.088|2.196|
> |TimeXL|**4.161**|**2.844**|**2.035**|
>
> Please kindly refer to our response to **reviewer g9bw** for more experimental results (or [this link](https://anonymous.4open.science/r/rebuttal-D4F5/Regression_results.pdf)) and discussions. We would also like to emphasize that we focus on classification-based prediction as many real-world multi-modal applications naturally involve discrete decision-making. This formulation also better highlights the interpretability benefits of our case-based explanations and the LLM’s reasoning capabilities. As for the baselines, TimesNet, OFA, TimeCAP inherently support classification, and the commonly-used TSLib (Appendix A.2) also uses most methods for classification. We also acknowledged regression tasks and provided results in Appendix F (Pages 23-25), including initial results and method designs.
>
>
> **Q2: Encoder choice**
>
> As discussed in Section 3.2.1 (Page 3), the model choice affects the explanation granularity, and we used convolutional neural networks followed by prototype layers for both modalities to capture segment-level prototypes for prediction and explanation.
>
> **Q4: Iterative process**
>
> The iterative process relies on training supervision to improve the text quality of training, validation, and testing sets. As detailed in Section 3.3.2 (Pages 5-6), TimeXL iteratively generates reflective feedback (by reasoning on text, prediction, and ground truths) to refine training texts, where validation data evaluates feedback quality per iteration. Feedback with the best performance is then applied to refine testing texts, which mimics applying a trained model to testing data (Page 6, Lines 284-288).
>
> **Q3 and Weakness 1 & 3:**
>
> **Refined input from LLMs**: In table 2, the ablation study on testing data is based on the refined texts, where the refinement is guided by reflective feedback from the best iteration selected by validation data (Page 6, lines 284-288).
>
> **Text and prototype only used for ablation study of LLM, not encoder**: We clarify that the prototypes are outputs of the encoder (Section 3.2.2, Prototype Projection, Page 4). As predictions and explanations cannot be derived without these prototypes, the ablation is not applicable. However, text and prototypes can both be input to the prediction LLM, and we perform such input ablation to show the performance gains from the contextual guidance provided by prototypes (Section 4.6, Page 8).
>
> **Importance of LLM components**: We show the importance of reflection and refinement LLMs via iterative analysis in Figure 5, Section 4.5. The original performance means no usage of reflection and refinement LLMs. It is clear that the text quality improves over the iterative reflection and refinements (upper two subplots). TimeXL’s prediction also improves (lower two subplots), because of the improved texts. The importance of prediction LLM is shown in Table 2. The multi-modal encoder indicates no usage of prediction LLM. After fusing with prediction LLM (text+prototype), its performance further improves (TimeXL). Appendix B includes more ablations .
>
> **Reference Discussion**
>
> Due to space limits, we summarized the insights of these works in the related work due to their differences from ours. News to Forecast constructs a large news database for method development and performs text-to-text prediction by fine-tuning LLM with selected news. Our method uniquely provides *explicit explanations* (standard case-based framework) from *both modalities*, showing key time series and text segments that contribute to the prediction. Besides, we emphasize the mutually augmented prediction between the multi-modal time series model and LLM for better performance. The most recently arxived ChatTime and ChatTS differ in scope: ChatTime examines modal translation capability, targeting zero-shot forecasting on common benchmarks, and reasoning tasks. ChatTS targets time series understanding and reasoning tasks instead of prediction. However, we thank the reviewer for pointing them out, and will provide detailed discussions in the updated version.

---

> > ### Comment · Reviewer_KxU3 · 2025-04-03
> >
> > Thanks for the rebuttal, which addressed many of my concerns. And I would like to raise my score.

---

### Decision · Program_Chairs · 2025-05-01

**Decision:**

Reject

**Comment:**

This paper introduces an architecture for classifying multi-modal time series/text data that builds on prototype learning to encode input time series and text to multi-modal prototypes and then passes prototypical text to LLMs to refine that inform final predictions that are an ensemble of predictions from the LLM and from the encoder. The result is a state-of-the-art classifier that involves intermediate text-generation steps that users could read to potentially inform their trust of the system.

Reviewers overall leaned slightly positive after the rebuttal phase but all had consistent reservations that should be addressed to improve the quality of the paper. They generally appreciated the introduction of having LLMs refine learned prototypes for new inputs and the fact that the proposed method appears to outperform some alternatives. Important reservations include some limits to the experiments, especially in explaining the value of the LLM text refinement (it appears the multi-modal encoder is doing most of the work) and despite the focus on explainability, the evaluation of explanations is small-scale and qualitative.